# Assessment of visually guided reaching in prodromal Alzheimer's disease: a cross-sectional study protocol

Alexandra G Mitchell ,[1] Robert D McIntosh,[1] Stephanie Rossit,[2] Michael Hornberger,[3] Suvankar Pal[4]

[1]School of Psychology, Philosophy & Language Sciences, The University of Edinburgh, Edinburgh, UK
[2]School of Psychology, University of East Anglia, Norwich, UK
[3]School of Medicine, University of East Anglia, Norwich, UK
[4]Anne Rowling Regenerative Neurology Clinic, Centre for Clinical Brain Sciences, The University of Edinburgh, Edinburgh, UK

**Correspondence to**
Dr Alexandra G Mitchell;
alexandra.mitchell@ed.ac.uk

## ABSTRACT

**Introduction** Recent evidence has implicated the precuneus of the medial parietal lobe as one of the first brain areas to show pathological changes in Alzheimer's disease (AD). Damage to the precuneus through focal brain injury is associated with impaired visually guided reaching, particularly for objects in peripheral vision. This raises the hypothesis that peripheral misreaching may be detectable in patients with prodromal AD. The aim of this study is to assess the frequency and severity of peripheral misreaching in patients with mild cognitive impairment (MCI) and AD.

**Methods and analysis** Patients presenting with amnestic MCI, mild-to-moderate AD and healthy older-adult controls will be tested (target N=24 per group). Peripheral misreaching will be assessed using two set-ups: a tablet-based task of lateral reaching and motion-tracked radial reaching (in depth). There are two versions of each task, one where participants can look directly at targets (free reaching), another wheren they must maintain central fixation (peripheral reaching). All tasks will be conducted first on their dominant, and then their non-dominant side. For each combination of task and side, a Peripheral Misreaching Index (PMI) will be calculated as the increase in absolute reaching error between free and peripheral reaching. Each patient will be classified as showing peripheral misreaching if their PMI is significantly abnormal, by comparison to control performance, on either side of space. We will then test whether the frequency of peripheral misreaching exceeds the chance level in each patient group and compare the overall severity of misreaching between groups.

**Ethics and dissemination** Ethical approval was provided by the National Health Service (NHS) East of England, Cambridge Central Research Ethics Committee (REC 19/EE/0170). The results of this study will be published in a peer-reviewed journal and presented at academic conferences.

## INTRODUCTION

The pathophysiological cascade that leads to Alzheimer's disease (AD) can begin up to 20 years before the onset of cognitive problems in both autosomal and sporadic AD.[1–5] In dominant and early-onset cases, there is evidence that the precuneus is one of the earliest regions to be affected.[5 6] Focal

---

### Strengths and limitations of this study

► The first study to systematically assess visually guided reaching in patients with cognitive impairment.
► Includes a simple tablet-based task (lateral reaching) that could be readily translated to clinical settings to assess the presence of peripheral misreaching.
► Case–control statistical tests of deficit are inherently low powered, so subtle deficits of reaching may not be detected at the level of individual patients.

---

damage in and around this brain area is known to be associated with deficits of visually guided action.[7] One example of such a condition is optic ataxia, an impairment of misreaching typically reflected in peripheral vision.[8 9] Patients with optic ataxia do not often report this symptom and it is rarely assessed in clinical settings, and it can therefore go undetected.[10] The changes observed in the precuneus in prodromal AD, and the link between the precuneus and optic ataxia, raise the hypothesis that optic ataxic misreaching may be detectable in patients with prodromal AD.

### Specific hypothesis

The hypothesis that peripheral misreaching is a feature of AD means that individual patients with AD, and possibly those with mild cognitive impairment (MCI), may show an abnormally large inflation of reaching errors when aiming for targets in peripheral vision, as compared with targets in free vision. At a group level, patients with AD and, to a lesser extent, patients with MCI may show significantly greater peripheral misreaching than healthy controls (HCs).

## METHODS
### Study setting

The study is a collaboration between clinicians and University staff at the University of

Edinburgh (UoE) and University of East Anglia (UEA). The details of recruitment and site information can be found in the online supplementary materials. Data collection for this study began on 03 October 2019 and 8 of 48 patients have taken part. Data for HCs have already been collected.

## Participants

Patients with a diagnosis of amnestic MCI or typical (amnestic) mild-to-moderate AD will be invited to take part. To ensure mild-to-moderate cases of AD, patients will have a score >50 in the most recent administration of Addenbrooke's Cognitive Examination (ACE-III).[11] If there is no recorded ACE-III score, clinical opinion of patient's condition will be used to assess eligibility.

Older adults without any known neurological disorders will be tested as an HC group. To achieve our target of 24 full datasets per group (Power considerations section), we plan to test up to 30 participants in each group, allowing for possible withdrawals.

### Inclusion criteria

For all participant groups, the ability to give informed consent is the initial inclusion criterion. Additional inclusion criteria are then applied to each group.

Control group inclusion criteria:
► Aged 50–75. (NB. The age-range for controls is targeted at the expected age range for patients, but the allowable range of ages for patients is wider than this, in order not to restrict recruitment unnecessarily.)
► No reported neurological or neurodegenerative conditions.

MCI group inclusion criteria:
► Aged 45–85.
► Clinical diagnosis of MCI with an amnestic pattern of presentation. This includes an observed deficit on cognitive/neuropsychological testing suggesting amnestic and visuospatial profile deficit, low β-amyloid, elevated phosphorylated Tau, regional atrophy on magnetic resonance brain imaging and/or regional perfusion changes on HMPAO-SPECT (Single photon emission computed tomography with hexamethyl propylenamine oxime).

AD group inclusion criteria:
► Aged 45–85.
► Clinical diagnosis of AD.

### Exclusion criteria

For all participant groups, the following exclusion criteria are applied:
► Significant difficulty communicating or understanding instructions in English.
► Significant, uncorrected visual impairment (eg, cataract, macular degeneration, scotoma, amblyopia and strabismus).
► Conditions that could interfere with smooth hand movements (eg, ataxia, essential tremor and severe arthritis).

► Clinical features suggestive of Lewy body pathology (eg, visual hallucinations or rapid eye-movement (REM) sleep disorder).

## Public and patient involvement

Patients with MCI or AD and their carers were involved in the early stages of planning and development. A focus group was held at the Anne Rowling Clinic in Edinburgh where patients and carers had the opportunity to try out prototypes of the tablet-based reaching task and provide feedback on task design. This feedback was used to optimise the final task for patient accessibility and clarity.

## Tasks

Two different set-ups will be used to assess peripheral reaching: a tablet-based assessment of reaching in the frontoparallel plane (lateral reaching), and a motion-tracking assessment of reaching in radial depth (radial reaching). Participants will complete two versions of each reaching task: a version in which participants look directly at targets before reaching to them (free reaching); and a version where central fixation is maintained (peripheral reaching). Any general factors affecting motor accuracy should influence both free and peripheral reaching, so we will treat the free reaching condition as a baseline condition, to be subtracted from peripheral reaching performance, to isolate the specific increase in error due to peripheral target presentation.[12] The critical outcome measure is therefore the inflation of absolute reaching error in peripheral reaching relative to free reaching.

Before testing, the participant's dominant writing hand is identified (by self-report). All tasks are completed first on the dominant side, using the dominant hand, followed by the non-dominant side and hand. Lateral reaching is completed first, followed by radial reaching. All tasks are performed in the same order for all participants.

### Lateral reaching tasks
#### Stimuli and apparatus

Stimuli are presented on an HP Pavillion ×360 touch screen (active display 310×175 mm, resolution 1920×1080 pixels). Tasks are controlled by a custom programme written in OpenSesame V.3.2.8 *Kafkesque Koffka*.[13] Participants are seated 40 cm away from the screen which is positioned with either the right edge (left-sided reaching, figure 1A) or the left edge (right-sided reaching, figure 1B) of the screen aligned to their midline figure 1A figure 1B. A start box (white rectangle, 2°×2°, 13.96×13.96 mm) is drawn at the edge (right or left) of the screen, aligned to participant's midpoint. In some tasks (detailed below) a white fixation cross is present (1°×1°, 6.98×6.98 mm), located 34.9 mm (5°) directly above the start box. Targets are white circles (diameter=2°, 13.96 mm) presented along radial spokes at 28°, 33° and 38° to the left (figure 1A) or right (figure 1B) of fixation. The experimenter sits across the table and monitors eye movements directly.

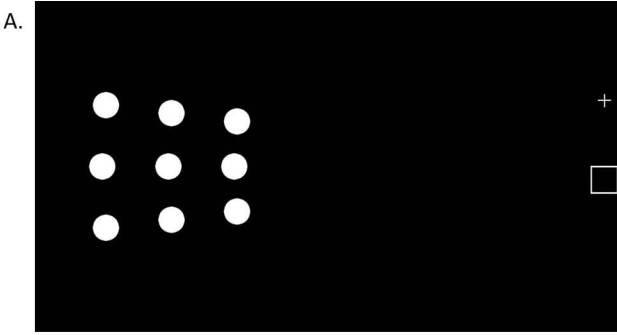

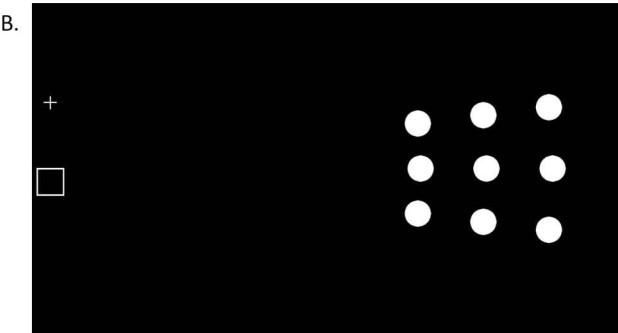

**Figure 1** Nine target positions for the lateral reaching task for left (A) and right (B) hand sides. At a viewing distance of 40 cm targets are presented at approximately 28°, 33° and 38° of eccentricity.

### Free reaching

For the first block in the lateral reaching task participants are not required to fixate, therefore, the fixation cross is absent.

Participants initiate a trial by pressing and holding down the start box, which disappears at touch. At this point, they may search the screen for a target. After a short delay (250–750 ms, randomised 100 ms intervals), a target appears at one of nine possible locations. As soon as the target appears, participants look at it and make one smooth reach to try to touch the target. The target remains on the screen until a touch is recorded at any location, and then the target disappears and a short beep (100 ms, 440 Hz) is played. The validity of the trial is then coded by the experimenter using a keyboard; 'y'—valid trial, 'e'—the participant did not move their eyes to the target, 'v'—void trial, and the start box reappears to begin another trial.

If an 'e' or 'v' is pressed, the corresponding trial is repeated until a valid trial is recorded. The block ends after a minimum of 27 valid trials (3 per target location), or after 50 valid and 'no eye-movement' trials.

### Visual detection

This is a simple check to confirm that the participant is capable of detecting the targets when presented in peripheral vision, to be allowed for a meaningful test of peripheral reaching (Peripheral reaching section).

Throughout each trial the participant must gaze at the fixation cross. They initiate the trial by pressing and holding down the start box, which disappears when

touched. In order to aid the maintenance of fixation, the fixation cross cycles between white and red at the screen refresh rate (60 Hz). After a short delay (250–750 ms), a target can appear at one of the nine locations for 1 s, or no target appears. This is followed by a short beep (100 ms, 440 Hz) to indicate the end of the trial. The participant must verbally report whether or not a target was seen in that interval. The experimenter records the response using the keyboard ('y'—yes, 'n'—no, 'v' - void). If the participant makes an eye movement, the experimenter presses 'e' and the trial is repeated. The block ends after 15 valid (no eye-movement) trials, one for each of the nine target locations, and six catch trials with no target.

To progress to the peripheral reaching task, participants are required to detect at least 6/9 targets and correctly rejects at least 3/6 catch trials. Otherwise, testing is discontinued on that side of space.

### Peripheral reaching

For peripheral reaching, participants are required to gaze at the fixation cross throughout each trial. A trial begins by pressing and holding down the start box. When touched, the start box disappears and the fixation cross cycles between white and red (at a rate of 60 Hz) until the trial ends. After a short delay (250–750 ms) a target appears at one of nine locations. While maintaining fixation, participants make one smooth reaching movement to try to touch the target. The target remains on the screen until a touch is recorded at any location, and a short beep is played once the target disappears. The experimenter then records the validity of the trial; 'y'—valid, 'e'—participant moved their eyes away from fixation, 'v'—void trial.

Invalid ('e' or 'v') trials are repeated until a valid trial is recorded. The block ends after a minimum of 27 valid trials (three per target location), or after 50 valid and 'eye-movement' trials.

### Radial reaching tasks
### Stimulus and Apparatus

An infrared motion-tracking camera (Optotrak Certus, Northern Digital) is used to track the reaching movement. Infrared-emitting diodes (IREDs) are taped to the tip of the right and left index fingers of each participant to track the reach in each hand. The Optotrak samples the IRED's 3D position at 100 Hz throughout each 2000 ms trial. The task is controlled by custom software written in LabView (National Instruments) programming environment. The stimuli and apparatus reported here are specific to UoE. At the second site, UEA, motion tracking was performed by using a Qualisys system (Gothenburg, Sweden) and a slightly different set up was used.[14]

Participants are seated with their head placed in a chin-rest in line with the middle of the display. Stimuli are back-projected via a mirror onto a screen (1000 mm wide × 750 mm deep) that lies flat in-front of the participant. A webcam is placed on the screen 50 cm directly in-front of the participant, as a fixation point. The live

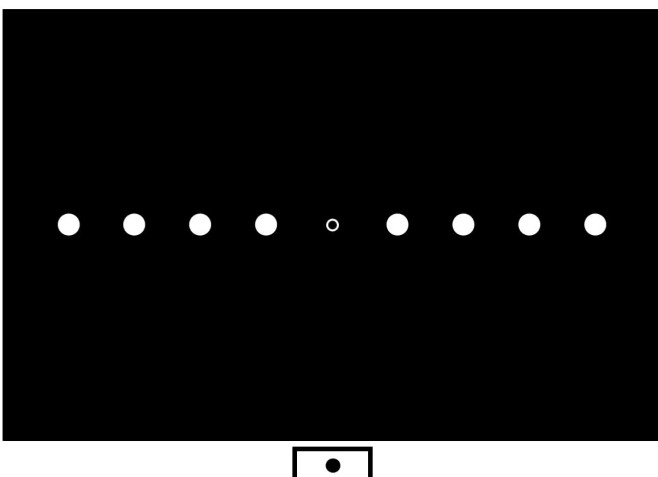

**Figure 2** Target positions for the radial reaching task shown here on both the right-hand and left-hand sides, at 11.4°, 22.6°, 33.4° and 43.6° from fixation. The start button is positioned at the bottom of the screen 40 cm away from central fixation. A webcam is placed at the point of central fixation (midpoint).

webcam image feeds into a separate laptop, allowing the experimenter to monitor gaze. A start button is aligned to the centre of the screen, positioned 10 cm in-front of the participant, 40 cm away from fixation (figure 2). Targets are white circles (diameter=1.60°, 13.96 mm) presented at four eccentric locations (11.4°, 22.6°, 33.4° and 43.6° away from centre) on each side (figure 2).

### Calibration

A calibration procedure is carried out before the reaching tasks to record the IRED position at the actual target location. A target is displayed at one target location and the participant is instructed to cover it completely with their reaching finger. Once the target is covered, the experimenter presses the start button and the finger location is recorded for 2000 ms. A beep plays after 2000 ms, indicating that the participant can move their hand away from the target position. Another target appears at the next location and the same procedure is repeated. Calibration is run using the ipsilateral hand for four targets on the left side and four on the right.

### Free reaching

Participants initiate a trial by pressing and holding down the start button. As soon as they push the button down, participants may look around the screen for a target. After 250–750 ms a target appears, participants then look directly at the target and reach to touch the target in one smooth movement. Optotrak recording is initiated simultaneously with target appearance, and the target disappearance is simultaneous with the end of the recording after 2000 ms. When the target disappears a short beep (100 ms, 440 Hz) plays, the participant leaves their finger at its landing position until they hear the beep. After the trial, the experimenter codes the trial validity with a key-press; 'Return'—valid, 'F1'—no eye-movement,

'Esc'—void trial. If an invalid trial ('F1' or 'Esc') is coded the trial gets recycled to the end of the block.

The block ends once 28 valid trials (7 per target location) are recorded, or after 50 valid and 'no eye-movement' trials.

### Peripheral reaching

To assess reaching accuracy in the periphery participants are required to look directly at central fixation (the webcam) throughout each trial. Participants initiate a trial by pressing and holding down the start button. After 250–750 ms a target appears. While maintaining gaze on the webcam participants make one smooth reaching movement to try to touch the target. After the reach, participants leave their finger at its landing position until a short beep (100 ms, 440 Hz). The target remains on screen for 2000 ms after the trial begins. The motion-tracker records the reach throughout the 2000 ms trial. At the end of the trial, the experimenter codes trial validity; 'Return'—valid trial, 'F1'—eye movement during trial, 'Esc'—void trial. If an invalid trial ('F1, 'Esc') is recorded then the trial is recycled to the end of the block.

The block ends after 28 valid trials (7 per target location) are recorded, or after 50 valid and 'eye-movement' trials.

## ANALYSIS PLAN
### Lateral reaching task

For the critical analyses, a single measure of reaching accuracy is taken for each participant, for each combination of viewing condition (free, peripheral) and side (non-dominant, dominant). For each response, the absolute error (in mm, x-axis and y-axis) is recorded as the distance of the reach endpoint from the target midpoint. The median absolute error is then calculated for each target eccentricity, across responses to the three targets at that eccentricity, for each combination of viewing condition and side. The average absolute error is then calculated as the mean of the medians for the three eccentricities, to give the single measure of reaching accuracy for each viewing condition at each side.

For the comparison of individual patients against control performance, the data are further compressed to a single index of performance per side, by subtracting reaching accuracy in the free vision condition from that in the peripheral condition. We call this value the Peripheral Misreaching Index (PMI).

### Analysis of single-case deficits

We will screen the control group for outliers that might suggest abnormalities, as such values would reduce the (already low, see figure 3) power to detect single-case deficits. We will use a robust method of outlier detection based on the median absolute deviation (MAD). The MAD can be multiplied by the consistency constant 1.4826 to estimate the SD, assuming a normal distribution. Each control participant's PMI can be expressed as

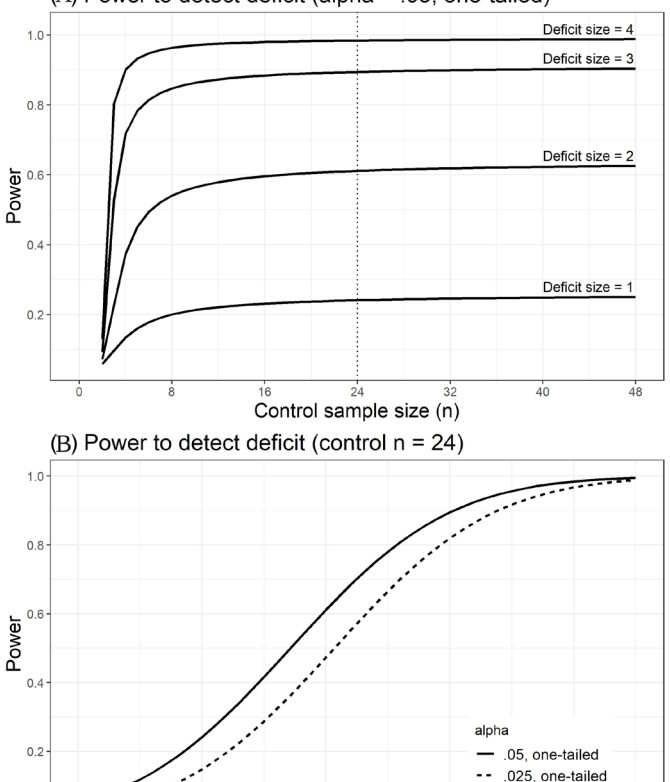

**Figure 3** Relation between(A) control sample size and power to detect a single-case deficit in a one-tailed test, for different sizes of deficit (D, expressed as SD of control mean). (B) D and power to detect a deficit, given a control sample size of 24, for adjusted (.025) and unadjusted (.05) alpha criteria.

a modified Z-score (Z′) by subtracting the group median, divided by the MAD *1.4826. If Z′ exceeds 2.5 on either side, that participant will be excluded, and replaced. Our simulations suggest that, for a group size of 24, we would expect to exclude (on average)<1 participant (~0.67) by this criterion.

We will next assess, for each side, whether the PMI of controls is related to age or sex, by computing Pearson's correlations. If the correlation is ≥0.3 on either side, then that variable will be included as a covariate in the subsequent case–control comparisons for both sides.

Case–control comparisons will then be run to compare each patient's PMI against control performance on each side of space. These comparisons will be based on Crawford and Howell's[15] modified t-test; or, if covariates are included, we will use the Bayesian Test of Deficit with covariates.[16] The individual tests will be one-tailed, with an alpha level set to 0.025, in order to constrain per-patient alpha level (across the two sides) to 0.05. If a patient shows a deficit on either side that meets the adjusted criterion (0.025), they will be classified as showing periperheral misreaching. If a patient shows a deficit on either side that would meet the unadjusted criterion (0.05), but not

the adjusted criterion, they will be classified as showing borderline peripheral misreaching.

Finally, a binomial test will test whether the rate of observed peripheral misreaching exceeds the rate expected by chance (ie, the per-patient adjusted alpha level of 0.05). A significant outcome (p<0.05) for either patient group will indicate that peripheral misreaching is a feature of this patient group. The observed rate of peripheral misreaching will provide an estimate of how common it is. We will run a further analysis including borderline cases and compare the rate of peripheral misreaching in each patient group against the appropriate chance level of 0.10.

### Group-level analysis
The case–control approach will be complemented by a group-level analysis of variance (ANOVA) of reaching accuracy, as measured by the PMI, with the between-subject factor of group (HC, MCI, AD) and the within-subject factor of side (non-dominant, dominant). This analysis will test whether the average severity of peripheral misreaching in each patient group significantly exceeds that observed in HCs.

### Exploratory analyses
Any lateralisation that occurs in MCI/AD is likely to be limited, therefore, any impairment in peripheral reaching may be similarly non-lateralised. An average PMI (across both sides) will therefore be calculated to assess peripheral reaching ability overall. More detailed analyses will be run with a between-subject factor of group and within-subject factors of side, eccentricity and viewing condition. These analyses will be conducted using dependent variables of absolute reaching error, directional (signed) reaching error, reaction time and movement time. The expectation is that peripheral misreaching will manifest as a fixation-directed bias, which is exacerbated at higher eccentricities significantly more so in patient groups than in age-matched controls.

### Radial reaching task
IRED speed is used to determine onset and offset of the reaching movement. Movement onset is defined as the first frame in which the IRED's speed exceeds 50 mm/s (and maintains that speed for up to 100 ms). Movement offset is defined as the first subsequent frame in which IRED speed falls below 50 mm/s. The landing position of the movement is defined by the x-coordinate and y-coordinate in the final frame of the movement and will be recorded as errors relative to true target locations recorded during calibration for each participant.

An initial analysis of PMI for the radial reaching task will be performed, restricted to the two most eccentric target positions (33.4° and 43.6°). Case–control comparisons follow the plan for the lateral reaching task (Analysis of single case deficits section), to estimate the rates of peripheral misreaching, and borderline peripheral misreaching, in the two patient groups. Due to different

experimental set-ups between the two test sites (UoE, UEA), each patient will be referenced to the same-site control data for case–control comparisons.

A group level ANOVA of PMI, restricted to the two most eccentric target positions, will similarly follow the plan for lateral reaching (Group-level analysis section). We will include site (UoE, UEA) as an additional covariate. More detailed analyses will also follow the plan for lateral reaching. Since motion tracking also provides kinematic variables on reaching trajectories, we also aim to examine the dependent variables peak speed and time to peak speed, normalised time after peak speed until reach endpoint and number of secondary movements.

## Power considerations

The target sample sizes (N=24 per group) are based on power considerations related to the main inferential analyses, the case–control comparisons and binomial tests of rates of peripheral misreaching deficits for the lateral reaching task.

The control sample size of 24 will provide close to the maximum power for case–control tests of deficit (figure 3A). Note that high power for these comparisons is inherently unachievable unless the deficit being tested for is very large. We do not know how large any misreaching deficits may be in our patient groups, but our control sample size will provide close to the maximum achievable power to detect them if they exist. Figure 3B illustrates more fully the relationship between deficit size (D) and power, for the adjusted alpha level (0.025) and unadjusted alpha level (0.05) by which we will determine peripheral misreaching deficits and borderline cases, respectively (Analysis of single case deficits section).

The main hypothesis is that peripheral misreaching will be found in a significant proportion of patients with AD and MCI. For one-sample binomial test to determine whether the observed rate of peripheral misreaching exceeds the chance level of 0.05, a sample size of 24 has >0.9 power. Provided that the true population proportion is at least 0.2 (1 in 5). This is appropriate to our aims, since peripheral misreaching would be of limited significance in these clinical populations if its prevalence were less than 1 in 5.

## ETHICS AND DISSEMINATION

This protocol was approved by UK Health Research Authority, by the East of England, Cambridge Central Research Ethics Committee on 13 June 2019 (REC reference 19/EE/0170).

All patients will provide informed consent, highlighting the voluntary nature of the study and their right to withdraw. If there is any doubt about the ability of the patient to provide informed consent, then this patient will not be recruited. There are no direct risks associated with taking part.

Careful consideration will be taken to maintain patient's confidentially. After consent is provided, an anonymous code will be assigned to each patient. Some patient details, such as the Community Health Index (CHI) number, age, gender and time of diagnosis, will need to be accessed by the research team. These details will be stored alongside patient code in a password-protected document.

At the end of the study, a lay summary of results will be provided to patients who have expressed a further interest. Project results will be made publicly available on the Open Science Framework (https://osf.io/bxnqs/) within 3 months after study end date (30 June 2020). Alongside this, we plan to publish the results of this protocol in a peer-reviewed journal and at academic conferences.

**Contributors** Each author has contributed significantly one or more aspects of the study. All authors contributed to study development and design. RDM, SR and AGM were involved in implementation of study protocol and analysis design. All authors contributed to data acquisition for mild cognitive impairment and alzheimer's disease cases, with SP and MH leading patient recruitment. AGM and SR were involved in data acquisition for HC. AGM and RDM drafted the manuscript and all authors provided critical revisions and approved the final version.

**Funding** This research is funded by the Dunhill Medical Trust Research Project Grant awarded to Prof Robert McIntosh & Dr Stephanie Rossit (RPGF1810\86).

**Competing interests** None declared.

**Patient consent for publication** Not required.

**Provenance and peer review** Not commissioned; externally peer reviewed.

**ORCID iD**
Alexandra G Mitchell http://orcid.org/0000-0001-8521-1891

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
