## [Reviewer comments · BMJ Open]

ARTICLE DETAILS

TITLE (PROVISIONAL)	The assessment of visually guided reaching in prodromal Alzheimer's disease: a cross-sectional study protocol
AUTHORS	Mitchell, Alexandra G; McIntosh, Robert D; Rossit, Stephanie; Hornberger, M; Pal, Suvankar

VERSION 1 – REVIEW

REVIEWER	Keir Yong UCL UK
REVIEW RETURNED	08-Dec-2019

GENERAL COMMENTS	This is a detailed and transparent study protocol for assessing visually guided reaching in patients with mild cognitive impairment and early to intermediate stage Alzheimer's disease (AD). Strengths include the clear rationale and interpretability of procedures and measures such as the peripheral misreaching index, as well as the exploratory analyses including directional error, which might be disproportionately evident in patient groups at greater eccentricities. As mentioned below, I think a particular strength is the exploratory analyses section. I have two main points. The first relates to distinguishing between findings from previous literature regarding the presymptomatic AD period: between observed versus estimated findings (and in which groups: e.g. familial versus sporadic AD), and between findings versus proposed criteria. The second is a query regarding rationale for analysing PMI separately for both sides. #1 Introduction: line 66-67; In my view it is important to either a) reference original data papers if making this statement, or if b) using the current Dubois et al. 2014 reference, state that research diagnostic criteria emphasise/propose this long asymptomatic phase. If adopting a), I suggest drawing a distinction between sporadic and autosomal dominant AD, which also relates to the Gordon et al. (2018) citation. If wanting to reference amyloid deposition being estimated as being abnormal for years during an asymptomatic period in sporadic AD, Villemagne et al. Lancet Neurol. (2013) may be relevant (though the authors note that estimates are based off observations from a relatively short interval). If of interest, VUmc has conducted a number of studies relevant to vulnerability of precuneal regions, e.g. in younger onset sporadic AD, cognitively healthy adults with family history of dementia or prodromal AD (Moller et al., 2013; ten Kate et al., 2016; ten Kate et al., 2018).
--

	#2 I was unsure of the rationale for testing for PMI on both sides separately (P14, line 319-320), as I would anticipate both patient groups will be exhibiting diffuse and ultimately bilateral pathological involvement of temporoparietal cortices. Although there might be some limited degree of lateralization, do you expect this would be sufficient to result in peripheral misreaching evident in one, but not necessarily both sides? Are there advantages of the advocated approach (per-patient side PMI with adjusted alpha level) relative to an unadjusted alpha level (.05) to evaluate evidence of a deficit based on per-patient mean PMI (averaging over both sides)? Minor P4 line 62 could be readily translated to clinical - missing word after 'clinical'? P5 hypotheses: possibly those with MCI: could this be more defined, e.g. 'At a group level, patients with AD, and to a lesser extent patients with MCI, will show greater peripheral misreaching than healthy controls.' P6: Participants: Exclusion- while you likely have considered these, perhaps including examples of clinical features suggestive of lewy body pathology as exclusion criteria might be worthwhile. P9: Visual detection: How long is the visual target displayed for? Is the beep also 100ms as in the Free reaching task? P14; i.e. the per-patient adjusted alpha level of .05 – I missed why this is adjusted, and why the unadjusted alpha level is .10 (lines 326, 332). P14: Exploratory analyses: I think this section is very strong. Only query was whether the expectation of peripheral misreaching exhibiting a fixation-dependent bias was anticipated to be particularly apparent in patient relative to control groups? P15: average spatial trajectory of reach. – can you elaborate on this measure?
--	---

REVIEWER	Andreas Johnen University Hospital Münster, Clinic of Neurology
REVIEW RETURNED	06-Feb-2020

GENERAL COMMENTS	This is a study protocol for a pilot study (?) about an interesting and potentially clinically relevant topic. The study protocol is well written and the visual reaching experiment is nicely set up. I have some basic remarks, however that should be considered by the authors:  - The authors use the word prodromal AD at several points however, they plan to only include aMCI and AD patients. These are not prodromal cases in the latest conceptualizations of AD. I highly suggest to include some kind of biomarkers for AD and particularly for aMCI (due to AD) to the inclusion criteria as this is state of the art now. - With the pilot data the authors cannot really relate the potential deficits to the parietal cortex / precuneus yet I believe. As this is a new experimental task to tap into optic ataxia, I find this crucial.
--

	The authors need to relate the behavioral data to MRI or atrophy ratings for a sound interpretation in that sense. - There are several other potentially confounding sources to consider besides the basic visual control tasks that are already included. I suggest to add more control tasks into the study protocol: 1) cognitive and particularly language and executive tasks: how can you be certain that the participants clearly understand what to do and do not forget the instructions during the experiment? 2) motor control tasks: how can you be sure that patients are able to follow more simple tasks that require motor control (e.g., grooved pegboard test / 9-hole peg test etc.)
--	---

VERSION 1 – AUTHOR RESPONSE

Reviewer: 1

Reviewer Name

Keir Yong

Institution and Country

UCL UK

Please state any competing interests or state 'None declared':

None declared

Please leave your comments for the authors below

This is a detailed and transparent study protocol for assessing visually guided reaching in patients with mild cognitive impairment and early to intermediate stage Alzheimer's disease (AD). Strengths include the clear rationale and interpretability of procedures and measures such as the peripheral misreaching index, as well as the exploratory analyses including directional error, which might be disproportionately evident in patient groups at greater eccentricities. As mentioned below, I think a particular strength is the exploratory analyses section.

I have two main points. The first relates to distinguishing between findings from previous literature regarding the presymptomatic AD period: between observed versus estimated findings (and in which groups: e.g. familial versus sporadic AD), and between findings versus proposed criteria. The second is a query regarding rationale for analysing PMI separately for both sides.

#1 Introduction: line 66-67; In my view it is important to either a) reference original data papers if making this statement, or if b) using the current Dubois et al. 2014 reference, state that research diagnostic criteria emphasise/propose this long asymptomatic phase. If adopting a), I suggest drawing a distinction between sporadic and autosomal dominant AD, which also relates to the Gordon et al. (2018) citation. If wanting to reference amyloid deposition being estimated as being abnormal for years during an asymptomatic period in sporadic AD, Villemagne et al. Lancet Neurol. (2013) may be relevant (though the authors note that estimates are based off observations from a relatively short interval). If of interest, VUmc has conducted a number of studies relevant to vulnerability of precuneal regions, e.g. in younger onset sporadic AD, cognitively healthy adults with family history of dementia or prodromal AD (Moller et al., 2013; ten Kate et al., 2016; ten Kate et al., 2018).

Thank you for these very helpful suggestions. We have taken your advice and referenced the original papers and updated the literature review accordingly on lines 66-69.

#2 I was unsure of the rationale for testing for PMI on both sides separately (P14, line 319-320), as I would anticipate both patient groups will be exhibiting diffuse and ultimately bilateral pathological involvement of temporoparietal cortices. Although there might be some limited degree of lateralization, do you expect this would be sufficient to result in peripheral misreaching evident in one, but not necessarily both sides? Are there advantages of the advocated approach (per-patient side PMI with adjusted alpha level) relative to an unadjusted alpha level (.05) to evaluate evidence of a deficit based on per-patient mean PMI (averaging over both sides)?

This is a point we discussed at length at the design stage. We eventually decided to test PMI on both sides separately because even a limited degree of asymmetry may create a unilateral reaching deficit which could be masked by relatively normal reaching performance on the other side.

However, we do take your point that any substantial impairment in AD is likely to occur on both sides and we have therefore added a step into the exploratory analysis that tests overall PMI averaged across sides, with unadjusted alpha level (lines 349-352)

Minor

P4 line 62 could be readily translated to clinical - missing word after 'clinical'?

Thank you, this now reads: 'Includes a simple tablet-based task (lateral reaching) that could be readily translated to clinical settings'

P5 hypotheses: possibly those with MCI: could this be more defined, e.g. 'At a group level, patients with AD, and to a lesser extent patients with MCI, will show greater peripheral misreaching than healthy controls.'

This has been added to lines 82-83

P6: Participants: Exclusion- while you likely have considered these, perhaps including examples of clinical features suggestive of lewy body pathology as exclusion criteria might be worthwhile.

Participants were excluded if they had clinical features suggestive of Lewy body pathology including prominent extrapyramidal signs, visual hallucinations, and pre-motor features such as REM sleep disorder, or a DaT scan suggestive of dopaminergic insufficiency. We have added this to the exclusion criteria (lines 133-134)

P9: Visual detection: How long is the visual target displayed for? Is the beep also 100ms as in the Free reaching task?

On line 197 it states: 'a target can appear at one of the nine locations for one second'

The beep is played for the same length of time as for all other tasks – we have added this information explicitly on line 204

P14; i.e. the per-patient adjusted alpha level of .05 – I missed why this is adjusted, and why the unadjusted alpha level is .10 (lines 326, 332).

The text explaining this was rather unclear and so we have written this portion of the methods to make the rationale for adjusting alpha levels clearer (lines 327-331).

P14: Exploratory analyses: I think this section is very strong. Only query was whether the expectation of peripheral misreaching exhibiting a fixation-dependent bias was anticipated to be particularly

apparent in patient relative to control groups?

Yes, that is our expectation and we have clarified this on line 357

P15: average spatial trajectory of reach. – can you elaborate on this measure?

We agree that this inclusion is a bit vague and have edited the manuscript to include more specific analyses (lines 380-381).

Reviewer: 2

Reviewer Name

Andreas Johnen

Institution and Country

University Hospital Münster, Clinic of Neurology

Please state any competing interests or state 'None declared':

None declared

Please leave your comments for the authors below

This is a study protocol for a pilot study (?) about an interesting and potentially clinically relevant topic. The study protocol is well written and the visual reaching experiment is nicely set up. I have some basic remarks, however that should be considered by the authors:

- The authors use the word prodromal AD at several points however, they plan to only include aMCI and AD patients. These are not prodromal cases in the latest conceptualizations of AD. I highly suggest to include some kind of biomarkers for AD and particularly for aMCI (due to AD) to the inclusion criteria as this is state of the art now.

This is a good suggestion, as participant recruitment for this study has already begun, unfortunately we cannot change the recruitment criteria. The diagnosis of aMCI, however, was not clear so we have clarified this in the updated manuscript (lines 116-119)

- With the pilot data the authors cannot really relate the potential deficits to the parietal cortex / precuneus yet I believe. As this is a new experimental task to tap into optic ataxia, I find this crucial. The authors need to relate the behavioral data to MRI or atrophy ratings for a sound interpretation in that sense.

This is a fair comment but is beyond the intended scope of the present investigation.

Parietal/precuneal involvement in AD is part of the rationale for a behavioural assessment of peripheral reaching in these clinical groups. The present study is a pilot study that aims to test whether individuals with AD show peripheral misreaching behaviourally. If they do, then it will become relevant to further ask whether this can be related to parietal cortex/precuneus, but this is not part of the intended design for the present study. Direct assessment of precuneus function would make a clear follow-up study. We cannot guarantee the routine access to high quality imaging that would allow us to pre-register such follow-up analyses. We take your point on board, however, and where we can we will obtain scans for each patient and will directly compare results with precuneus in these patients if we obtain enough data.

- There are several other potentially confounding sources to consider besides the basic visual control tasks that are already included. I suggest to add more control tasks into the study protocol: 1) cognitive and particularly language and executive tasks: how can you be certain that the participants clearly understand what to do and do not forget the instructions during the experiment? 2) motor control tasks: how can you be sure that patients are able to follow more simple tasks that require motor control (e.g., grooved pegboard test / 9-hole peg test etc.)

These are good, very important considerations and we did discuss these issues at length during the design process. We believe that we have taken account of these factors adequately in the existing design, at least considering the practical constraints of the study.

1) The ability of each individual to understand and comprehend instructions is assessed at multiple stages. Firstly, only patients with an ACE-III score > 50 are recruited, to minimise likelihood that they will be unable to comprehend task demands. Before recruitment, the clinician decides whether patients that fit the study criteria are able to take part in the study, this includes their ability to follow and understand instructions. As a final measure, before taking consent, the experimenter makes the decision to continue their participation based on their comprehension and understanding of the study and tasks involved. If the patient is unable to complete the task at any point, the experimenter stops testing and their data is removed from final analysis.

2) The free reaching task is specifically included to provide a baseline control task which allows us to factor out any baseline motor impairment. We did not clearly spell out this rationale in the initial submission and have now made the role of the free vision condition explicit (lines 149-153)

VERSION 2 – REVIEW

REVIEWER	Keir Yong UCL UK
REVIEW RETURNED	02-Mar-2020

GENERAL COMMENTS	The authors have adequately addressed my concerns. The revised article is suitable for publication.
---

REVIEWER	Andreas Johnen Clinic of Neurology with Institute For Translational Neurology University Hospital Münster Germany
REVIEW RETURNED	27-Feb-2020

GENERAL COMMENTS	The authors have improved the previous version of the study protocol. All my concerns and comments have been sufficiently addressed.
--